# Detection of BRAF mutations in malignant melanoma and colorectal cancer by SensiScreen® FFPE BRAF qPCR assay

Anna Lahn Sørensen[1]*, Mariann Guldmann-Christensen[2], Michael Børgesen[1], Rasmus Koefoed Petersen[1], Katharina Flugt[1], Julie Mejer Holmgaard Duelund[2], Majbritt Hauge Kyneb[3], Jan Lorenzen[3], Emma Pipó-Ollé[1], Samantha Epistolio[4], Alice Riva[4¤], Giulia Dazio[4], Elisabetta Merlo[4], Tine Meyer[2], Ulf Bech Christensen[1], Milo Frattini[4]

**1** PentaBase A/S, Odense, Denmark, **2** Department of Pathology, Laboratory of Research and Development, Aarhus University Hospital, Aarhus, Denmark, **3** Danish Technological Institute, Aarhus, Denmark, **4** Institute of Pathology EOC, Ente Ospedaliero Cantonale, Locarno, Switzerland

¤ Current address: Clinical Chemistry and Immunometry Department, Ospedale Generale di Zona Moriggia Pelascini, Gravedona, Italy

* als@pentabase.com

**Data Availability Statement:** All relevant data are within the manuscript and its Supporting Information files.

## Abstract

Mutations in *BRAF* exon 15 lead to conformational changes in its activation loops, resulting in constitutively active BRAF proteins which are implicated in the development of several human cancer types. Different BRAF inhibitors have been developed and introduced in clinical practice. Identification of *BRAF* mutations influences the clinical evaluation, treatment, progression and for that reason a sensitive and specific identification of *BRAF* mutations is on request from the clinic. Here we present the SensiScreen® FFPE BRAF qPCR Assay that uses a novel real-time PCR-based method for *BRAF* mutation detection based on PentaBases proprietary DNA analogue technology designed to work on standard real-time PCR instruments. The SensiScreen® FFPE BRAF qPCR Assay displays high sensitivity, specificity, fast and easy-to-use. The SensiScreen® FFPE BRAF qPCR Assay was validated on two different FFPE tumour biopsy cohorts, one cohort included malignant melanoma patients previously analyzed by the Cobas® 4800 *BRAF* V600 Mutation Test, and one cohort from colorectal cancer patients previously analyzed by mutant-enriched PCR and direct sequencing. All *BRAF* mutant malignant melanoma patients were confirmed with the SensiScreen® FFPE BRAF qPCR Assay and additional four new mutations in the malignant melanoma cohort were identified. All the previously identified *BRAF* mutations in the colorectal cancer patients were confirmed, and additional three new mutations not identified with direct sequencing were detected. Also, one new *BRAF* mutation not previously identified with ME-PCR was found. Furthermore, the SensiScreen® FFPE BRAF qPCR Assay identified the specific change in the amino acid. The SensiScreen® FFPE BRAF qPCR Assay will contribute to a more specific, time and cost saving approach to better identify and characterize mutations in patients affected by cancer, and consequently permits a better *BRAF* characterization that is fundamental for therapy decision.

**Funding:** This work was supported by Innovation Fund Denmark and EU trough EUREKA- Eurostars grant E10440, in the form of salaries for ALS, MB, RKP, EPO, KF, MHK, JL and UBC. Eurostars (https://www.eurekanetwork.org/) did not have any additional role in the study design, data collection and analysis, decision to publish, or preparation of the manuscript.

**Competing interests:** ALS, MB, RKP, EPO, KF, MHK, JL and UBC received salaries from Eurostars. ALS, MB, RKP, EPO, KF and UBC are employees of PentaBase A/S. The funding organization did not play a role in the study design, data collection and analysis, decision to publish, or preparation of the manuscript and only provided financial support in the form of authors' salaries and/or research materials. SensiScreen FFPEBRAF qPCR is part of a marketed product portfolio of PentaBase A/S. This does not alter our adherence to PLOS ONE policies on sharing data and materials.

# 1. Introduction

In 2002 Davies and colleagues demonstrated that the occurrence of a specific mutation in exon 15, codon 600 (formerly identified as 599) can constitutively activate the BRAF (B-rapidly accelerated fibrosarcoma protein) [1]. Codon 600 encodes a part of the activation loop in the kinase domain of the BRAF protein, and mutations in this region result in a substitution of valine by different amino acids, such as glutamate (V600E), arginine (V600R), lysine (V600K) or aspartate (V600D). These substitutions, especially the V600E, can lead to an abnormal increase in the catalytic activity of the BRAF protein by inducing conformational changes or mimicking a change in the phosphorylation status in its activation loop and leading to a BRAF protein with constitutively active kinase activity. As a consequence of these mutations, the BRAF -MEK (Mitogen-activated protein kinase kinase)–ERK (Extracellular signal-regulated kinase)—pathway is constitutively active, leading to uncontrolled cell growth, cell division and subsequently cancer development [1–3]. Almost all mutations in the *BRAF* gene occur at the hotspot codon 600 and roughly 90% are represented by the V600E change. However, although rarer, non-V600E mutations like V600K, V600D and V600R substitutions are observed as well [1].

BRAF mutations have been described in about 50% of malignant melanoma (MM) [4], 50% of papillary thyroid carcinoma, 5–10% of colorectal cancer (CRC) [5], and, in a lesser extent, in lung adenocarcinomas about 2–4% of cases and low-grade glioma especially the pilocytic astrocytoma subtype [1, 6–12]. In MM and CRC, *BRAF* mutations play the most relevant clinical role. Even in lung adenocarcinoma *BRAF* mutations play a clinical role, although limited by the low frequency [13].

At the clinical level, the treatment of MM has greatly been improved by the development of selective inhibitors against *BRAF* mutations in codon 600 (e.g., vemurafenib) [14]. To initiate a treatment with targeted therapy in patients with stage IV metastatic melanoma the identification of a *BRAF* mutation in codon 600 is mandatory [15]. The Food and Drug Administration (FDA) approved BRAF inhibitors for patients harboring the *BRAF* V600E mutation only, while the European Medicine Agency (EMA) for all patients with a mutation in the same codon. The identification of the *BRAF* mutation is clinically relevant since these patients are more unlikely to undergo metastasectomy as the metastases are present at sites not assessable for surgery, usually peritoneal disease, or distant lymph node metastasis [16, 17]. Moreover, identification of the *BRAF* V600E mutation in patients with metastatic CRC is associated with a poor prognosis for obtaining modest benefit from treatments with chemotherapy, particularly in the second- and third-line settings compared with *BRAF* WT cases [3, 5, 16]. An approved treatment of CRC is administration of monoclonal antibodies against the epidermal growth factor receptor (EGFR) (e.g., cetuximab and panitumumab). These antibodies bind the extracellular domain on EGFR and inhibit ligand binding and dimerization, rendering the downstream signaling pathway unsusceptible for activation. It is believed that *BRAF* mutations are responsible for 12–15% of patients who fail anti-EGFR treatment, similarly to what demonstrated for *KRAS/NRAS* mutations. Therefore, it is recommended only to use anti-EGFR drugs in patients whose tumours display *BRAF/KRAS/NRAS* WT sequences [16, 18].

At present, there exist various methodologies to analyze the *BRAF* mutations. The cheapest is immunohistochemistry. However, this method is only able to identify the V600E change [19, 20]. Further methodologies include direct sequencing (DS), next-generation sequencing (NGS), mass spectrometry and real-time PCR based methods such as Cobas® BRAF mutation test (Roche), Therascreen BRAF PCR kit (Qiagen), and Idylla™ BRAF Mutation Test (Biocartis), overall characterized by marked differences in limit-of-detection (LOD): 1–10%, down to 5%, around 5%, 0.5–1% and 1%, respectively. Before choosing the methodology for *BRAF*

evaluation, handling time, time to answer, skills, the expertise of the technical personnel required for performing the analysis and finally, costs should be considered. All these aspects may have a significant impact in the clinical management of patients in terms of proper diagnosis, classification and, in the worst case, in terms of missing the opportunity of a targeted treatment.

In the present study, we report the development and testing of a real-time PCR assay based on a new technology, for the identification of mutations in the *BRAF* gene on formalin-fixed paraffin-embedded (FFPE) tissue, named SensiScreen® FFPE BRAF qPCR Assay. The SensiScreen® FFPE BRAF qPCR Assays are based on the PentaBase DNA analogue technology, where oligoes are modified with intercalator, also known as pentabase. Oligoes with pentabases are defined as intercalating nucleic acid (INA®). Electrons from the pentabase are participating in the stacking of DNA duplexes [21, 22]. The samples were tested by the SensiScreen® FFPE BRAF qPCR simplex and multiplex assays and the results compared to DS, mutant-enriched PCR (ME-PCR) or to the Cobas® 4800 *BRAF* V600 Mutation Test (Roche). The overall purpose of this study was to validate a highly sensitive, specific, short turnaround time, low cost and easy-to-use assay for *BRAF* characterization able to work on the most common real-time PCR instruments. Furthermore, the SensiScreen® FFPE BRAF qPCR Assay was developed to use a minimal input of DNA.

## 2. Material and methods

### 2.1. Cell lines

The cell lines used for the sensitivity studies contained four different mutations in *BRAF* codon 600 (V600E, V600K, V600D and V600R). The sensitivity tests were conducted using cell line DNA extracted from HT-29 (ATCC®; catalogue number: HTB-38™), IGR-1 (DSMZ®; catalogue number: ACC 236™), WM-115 (Rockland Immunochemicals®; catalogue number: WM115-01-0001™) and HCT116 *BRAF* (V600R/+) (Horizon Discovery Group®; catalogue number: HD 104-037™) (Table 1). Subcultures were made in appropriate media according to the manufacturer's instructions and genomic DNA was isolated using the QIAmp Mini kit (Qiagen, Chatsworth, CA, USA).

### 2.2. Sensitivity studies

LOD of the SensiScreen® FFPE BRAF qPCR Assay was evaluated by serial dilutions of cell line DNA, harboring *BRAF* mutations, in a background of 50 ng WT human genomic DNA (Promega, Madison, Wisconsin, USA). Five different concentrations of mutated DNA were tested (10%, 5%, 2%, 1% and 0.5%) on MyGo Pro real-time PCR instruments (IT-IS Life Science Ltd, Mahon, Ireland) and Rotor-Gene 6000 (Corbett Research), using the SensiScreen® protocol previously described [22]. The SensiScreen® FFPE BRAF qPCR Assay limit of blank (LOB) was evaluated by application of 20 replica of 50 ng WT human genomic DNA (Promega, Madison, Wisconsin, USA).

**Table 1. Features of the cell lines used for the sensitivity studies.**

| Cell line | Mutation | Zygosity | Origin |
|---|---|---|---|
| HT-29 (ATCC® HTB-38™) | Val600Glu (p.V600E, *c.1799T>A*) | Heterozygous | Colorectal cancer |
| IGR-1 (DSMZ® ACC-236™) | Val600Lys (p.V600K) | Heterozygous | Melanoma |
| WM115 (ATCC® CRL-1675™) | Val600Asp (p.V600D) | Heterozygous | Melanoma |
| HCT-116 (ATCC® CCL-247 ™) | Val600Arg (p.V600R) | Heterozygous | Colorectal cancer |

## 2.3. SensiScreen® FFPE BRAF qPCR assay

To develop a real-time PCR assay able to specifically amplify few copies of mutated DNA in a large WT background we took advantage of PentaBases intercalating nucleic acid technology. The development of the SensiScreen® FFPE BRAF qPCR Assay was performed as previously described [22].

## 2.4. Patient samples

Patients included in this study were diagnoses with MM (cohort recruited at the Department of Pathology, Aarhus University Hospital, Denmark) or with CRC (cohort recruited at the Institute of Pathology EOC, Locarno, Switzerland). Only patients with sufficient tumour tissue material have been included. The tumour materials used in the study were surplus to requirements for routine testing and were obtained after surgery and clinicopathological assessment of the tumour as part of the standard clinical procedure. All data and materials were fully anonymized before assessment. This study was approved by the Institutional Ethical Committee of the Institute of Pathology EOC of Locarno, Switzerland, by the Central Denmark Region Committee on Health Research Ethics, and by the Danish Data Protection Agency. All procedures were performed in accordance with the ethical standards of the Helsinki Declaration.

**2.4.1. Tissue analyses.** Quality and tumour cell content of the FFPE tissue block sections were evaluated by an experienced pathologist. Genomic DNA was extracted from representative FFPE tissue blocks, containing at least 50–70% neoplastic cells. In cases of low tumour tissue content, a macrodissection was performed to enrich the amount of tumour cells. Genomic DNA was extracted from a 5 μm thick section of MM tissue, using the Cobas® DNA Sample Preparation Kit (Roche) according to the manufacturer's instructions. In CRC samples, genomic DNA was extracted from six 7 μm thick serial FFPE sections. DNA extraction was performed using the QIAmp Mini kit (Qiagen) according to the manufacturer's instructions.

**2.4.2. Cohort 1.** The first cohort included 127 patients diagnosed with histologically confirmed MM collected from 2012 to 2013 at the Department of Pathology, Aarhus University Hospital, Denmark. *BRAF* exon 15 was retrospectively analyzed using the Cobas® 4800 *BRAF* V600 Mutation Test (Roche) and the SensiScreen® FFPE BRAF qPCR Simplex and Multiplex Assays (PentaBase). Four patients were excluded from both tests, due to insufficient DNA. Clinical-pathological features are included in S1 Table.

**2.4.3. Cohort 2.** The second cohort included 100 patients affected by histologically confirmed CRC and collected from 1996 to 2009 at the Institute of Pathology EOC of Locarno, Switzerland. The samples were retrospectively analyzed for *BRAF* exon 15 mutations by DS, ME-PCR and the SensiScreen® FFPE BRAF qPCR Simplex and Multiplex assays. Three patients were excluded from the SensiScreen® FFPE BRAF qPCR assay due to long term storage, resulting in insufficient and invalid DNA. Clinical-pathological features are included in S1 Table.

## 2.5. Mutational analysis

### 2.5.1. Direct sequencing

The DS analyses were performed at the Institute of Pathology EOC (Locarno, Switzerland) using the primers listed in Table 2. The samples were subjected to automated sequencing on an ABI PRISM 3130 Genetic Analyzer (Applied Biosystems, Foster City, CA, USA), and evaluated using the Sequencing Navigator Software (Applied Biosystems). The identified mutations were confirmed in two independent PCR reactions run on ProFlex PCR System (Thermo-Fisher Scientific, Waltham, MA, USA).

**Table 2. Sequences of oligonucleotides used for construction of model templates for DS, ME-PCR and SensiScreen assays.**

| Primer name | Primer sequence 5´-> 3´* |
|---|---|
| BRAF cloning forward | CTGTTTTCCTTTACTTACTACACCTC |
| BRAF cloning reverse | GTGGAAAAATAGCCTCAATTCTTAC |
| DS forward | TCATAATGCTTGCTCTGATAGGA |
| DS reverse | GGCCAAAAATTTAATCAGTGGA |
| ME-PCR forward (1st and 2nd PCR) | TAAAAATAGGTGATTTTGGTCTAGCTGC |
| ME-PCR reverse (1st PCR) | CCAAAAATTTAATCAGTGGAAAAATA |
| ME-PCR reverse (2nd PCR) | AAAAATTTAAGCAGTGGAAAAATAGC |
| SensiScreen® V600 reference forward | ATAGGTGATTTTGGTCTAGCTAC |
| SensiScreen® V600D forward | GTGATTTTGGTCTAGCTACAGAT |
| SensiScreen® V600E simplex forward | GGTGATTTTGGTCTAGCTACCGAG |
| SensiScreen® V600E multiplex forward | GGTGATTTTGGTCTAGCTACCGA |
| SensiScreen® V600K forward | GGTGATTTTGGTCTAGCTATAAA |
| SensiScreen® V600R forward | GGTGATTTTGGTCTAGCTATAAG |
| SensiScreen® reverse | GTAAGAATTGAGGCTATTTTTCCAC |
| SensiScreen® probe | *Penta Green™*-ATGGAGTGGGTCCCATCAGTTTG-*Green Quencher* |
| SensiScreen® WT blocker A | TCTAGCTACAGTGAAATCTC |
| SensiScreen® WT blocker B | TCTAGCTACAGTGAAATCTCGA |
| SensiScreen® V600D blocker | TCTAGCTACAGATAAATCTC |
| SensiScreen® V600E blocker | TCTAGCTACAGAGAAATCTC |
| SensiScreen® V600K blocker | TCTAGCTACAAAGAAATCTCGA |
| SensiScreen® V600R blocker | TCTAGCTACAAGGAAATCTCGA |
| SensiScreen internal control probe | *Penta Yellow™*-CTACTCCACTGCTGTCTAT-*Yellow Quencher* |
| SensiScreen internal control forward | CCCTAGAGTTGCCACAGC |
| SensiScreen internal control reverse | GGTAAGCAGCAAGAGAGC |

*Proprietary modifications not shown. Abbreviations; DS: Direct sequencing; ME-PCR: Mutant-enriched PCR; PCR: polymerase chain reaction; WT: Wild type.

**2.5.2. Mutant-enriched PCR.** Analysis of *BRAF* exon 15 mutations by ME-PCR was performed at the Institute of Pathology EOC (Locarno, Switzerland), essentially as described previously [15, 23, 24] ME-PCR allows the enrichment of mutant alleles and the elimination of WT alleles by application of a restriction enzyme digestion on the PCR amplification product [25–27]. The ME-PCR products were subsequently subjected to automated sequencing and analyzed as described above. All mutated cases were confirmed twice in independent PCR reactions.

**2.5.3. Cobas® BRAF mutation test.** This *BRAF* exon 15 mutations analysis was performed at the Department of Pathology at Aarhus University Hospital (Denmark) using the Cobas® 4800 *BRAF* V600 Mutation Test on a Cobas® 480Z analyzer. DNA extraction was conducted as previously described [22]. The DNA concentration was measured using an Implen Nanophotometer (Germany). Subsequently, the DNA was diluted to a concentration of 5 ng/μl in water. Twenty-five μl mixture was added in each reaction tube.

**2.5.4. SensiScreen® FFPE BRAF qPCR assay.** The SensiScreen® FFPE BRAF qPCR Simplex and Multiplex mutation analysis on MM and CRC were performed at the Department of Pathology, Aarhus University Hospital, Denmark and at the Institute of Pathology, Locarno, EOC, Switzerland, respectively.

Development of the SensiScreen® FFPE BRAF qPCR Assay for MM and CRC was performed on a Mx3005P qPCR system (Stratagene, CA, USA) and a CFX96 (Bio-Rad, Hercules, CA, USA), respectively. The reaction was conducted in 25 μL with Ampliqueen 2xMaster Mix

(PentaBase, Odense, Denmark), 300–900 nM of each SuPrimers, 200 nM of each Hydro-lEasy probes, and 1000–5000 nM WT BaseBlocker. All SensiScreen® FFPE BRAF qPCR Assays contained an internal control targeting the *CYP17A1* gene. The multiplex assay included SuPrimers for detection of V600E, V600D, V600K and V600R mutations in a single tube, and was used for the initial screening of *BRAF* mutations (Table 2). The individual V600 mutation was subsequently specified when using the simplex assay–each tube containing a single *BRAF* mutation. *In vitro* specificities were analyzed using 50 ng of cell line DNA or 50 ng human genomic WT DNA (Promega G304A) in the absence or presence of 1000 copies of mutated plasmid DNA carrying *BRAF* mutations. FFPE purified DNA was diluted to a concentration of 1–10 ng/ μL in water. Five μL was added to each reaction tube. The PCR program was: 2 min at 95˚C, followed by 45 cycles of 15 sec at 94˚C and 60 sec at 60˚C.

**2.5.5. Data processing.**   The threshold cycle (Ct) was defined as 10% of the signal strength (RFU) of the reference assay at cycle 45. The ΔCt value was calculated for each mutation specific analysis and defined as the difference between the Ct value from the given mutation analysis subtracted the Ct value from the corresponding reference analysis. ΔCt was calculated for all mutation assays having a Ct value beneath 39.

$$\mathbf{\Delta Ct = Ct_{mutation} - Ct_{reference}}$$

The ΔCt cut-off was defined as the value that will avoid false positive results from amplification of WT samples and at the same time identify as many true positive samples as possible. Since a ΔCt at 12.1 was the lowest value identified when using WT DNA as template for mutation assay, we considered a ΔCt of 9 as the optimal value for subsequent analysis of assay sensitivities. A sample was defined as positive for a given mutation if Ct of the mutation assay was beneath 39 and the ΔCt value beneath 9.

To determine the LOD, a serial dilution experiments were made in duplicate for each plasmid to achieve concentrations from 16.000 to 0 copies/reaction. The serial dilutions were used to establish a calibration curve for each mutated DNA. The LODs values were determined and calculated from the calibration curve, where *y* is $Ct_{reference}$ +9. This was determined for both simplex and multiplex. PCR amplification efficiency was determined from the slope of the log-linear portion of the calibration curve (Efficiency = $10^{-1/slope}$-1).

# 3. Results

## 3.1. SensiScreen® FFPE BRAF qPCR assay sensitivity

To evaluate the sensitivity of the SensiScreen® FFPE BRAF qPCR Simplex and Multiplex assays, a serial dilution of cell line DNA harboring *BRAF* mutations in a background of human genomic WT DNA was performed in duplicate. The LOD for SensiScreen® FFPE BRAF qPCR Simplex ranges from 0.55% to 1.04%, with V600R showing the lowest value (Table 3). Looking at the SensiScreen® FFPE BRAF qPCR Multiplex assay the range is from 0.66% to 1.81% with V600D and V600E showing the lowest value. V600E shows identical LODs in SensiScreen® FFPE BRAF qPCR Simplex and Multiplex. The equation for the dilution series curve for alle PCR assays showed that the PCR amplification efficiencies and correlation coefficient ($R^2$) were within the acceptable range. The efficiency ranged between 77% and 109%, and the $R^2$ values were all greater than 0.96. Calibration curves for each target are included in the S1 Fig. SensiScreen® FFPE BRAF qPCR Assays LOB was evaluated for each variant by 20 replicas of human genomic WT DNA. Mean ΔCt and standard deviation for each SensiScreen® FFPE BRAF qPCR is stated in Table 3.

**Table 3. Sensitivity and efficiency of SensiScreen® FFPE BRAF qPCR assay on mutated DNA in WT background.**

| Assay | LOD (%) | LOB | ΔCt WT | $R^2$ | Efficiency (%) | Equation |
|---|---|---|---|---|---|---|
| | | | **BRAF** | | | |
| V600D Spx | 0.68% | 16.54±0.25 | 12.1 | 0.997 | 91 | y = -3.55x + 40.39 |
| V600E Spx | 0.73% | 14.19±0.38 | 18.1 | 0.997 | 90 | y = -3.59x + 40.30 |
| V600K Spx | 1.04% | NA | No signal | 0.999 | 81 | y = -3.87x + 41.56 |
| V600R Spx | 0.55% | NA | No signal | 0.994 | 109 | y = -3.12x + 39.27 |
| V600D Mpx | 0.66 | 13.03 ±0.58 | No signal | 0.973 | 91 | y = -3.56x + 40.38 |
| V600E Mpx | 0.66 | | 17 | 0.966 | 93 | y = -3.50x + 40.09 |
| V600K Mpx | 1.81 | | No signal | 0.981 | 82 | y = -3.85x + 41.56 |
| V600R Mpx | 0.73 | | 15.5 | 0.98 | 77 | y = -4.05x + 41.45 |

Limit of detection (% mutation) was determined by the calibration curve for each mutant, where $y$ is Ct$_{reference}$ +9. Limit of blank was determined as mean ΔCt and standard deviation. Abbreviations; Ct: Threshold cycle; Efficiency: PCR amplification efficiency; LOD: Limit of detection; LOB: Limit of Blank; Mpx: Multiplex assay; ND: Not determined; $R^2$: Correlation coefficient; Spx: Simplex assay; WT: Wild type.

## 3.2. Clinical performance of the SensiScreen® FFPE BRAF qPCR assay

To validate the SensiScreen® FFPE BRAF qPCR Assay in a clinical setting, the assay was tested on DNA from FFPE tumour tissues from two different cohorts, S1 Table. In cohort 1, FFPE tumour biopsies from 127 Danish MM patients, previously analyzed by the Cobas® 4800 *BRAF* V600 Mutation Test were retrospectively analyzed by the SensiScreen® FFPE BRAF qPCR Simplex and Multiplex assays. In cohort 2, 100 FFPE tumour specimens from CRC patients previously analyzed by ME-PCR and DS were retrospectively tested by the SensiScreen® FFPE BRAF qPCR simplex Assay. In general, there was a good separation between the WT and the mutant group in both cohorts when using the SensiScreen® FFPE BRAF qPCR Assay analysis settings defined as positive for a given mutation if Ct of the mutation assay was beneath 39 and the ΔCt value beneath 9, Fig 1.

The SensiScreen® FFPE BRAF qPCR Assay confirmed the same mutations already identified by Cobas® 4800, ME-PCR and DS (Table 4). Furthermore, in cohort 1, the SensiScreen® FFPE BRAF qPCR Assays (both simplex and multiplex) identified four mutations not already identified with Cobas®, corresponding to an additional identification of 8% mutated cases at the MM cohort. Beside identification of not previously detected mutations, the SensiScreen® FFPE BRAF qPCR Assay also identified specific changes in amino acids, these variants were the V600K in two cases, and the V600E and the V600R in one case each (no V600D variants were identified). In cohort 2 the SensiScreen® FFPE BRAF qPCR Assay identified three new mutations not previously found with DS, all represented by a V600E change. Identification of the three new mutations corresponds to an additional 50% mutated cases in the CRC cohort not already identified with DS. The SensiScreen® FFPE BRAF qPCR Assay also identified a new mutation not previously identified with ME-PCR, corresponding to an additional 12.5% mutated cases in the CRC cohort not already identified with ME-PCR.

## 4. Discussion

A high sensitivity and specificity for detection and identification of V600 mutations in the *BRAF* gene are of great importance, since it has essential consequences for the choice of proper treatment of the patients. Indeed, the characterization of this genetic loci helps to predict the response to the treatment and consequently avoid the unnecessary administration of expensive therapies related to huge side effects and decreasing the quality of life of the patients. An

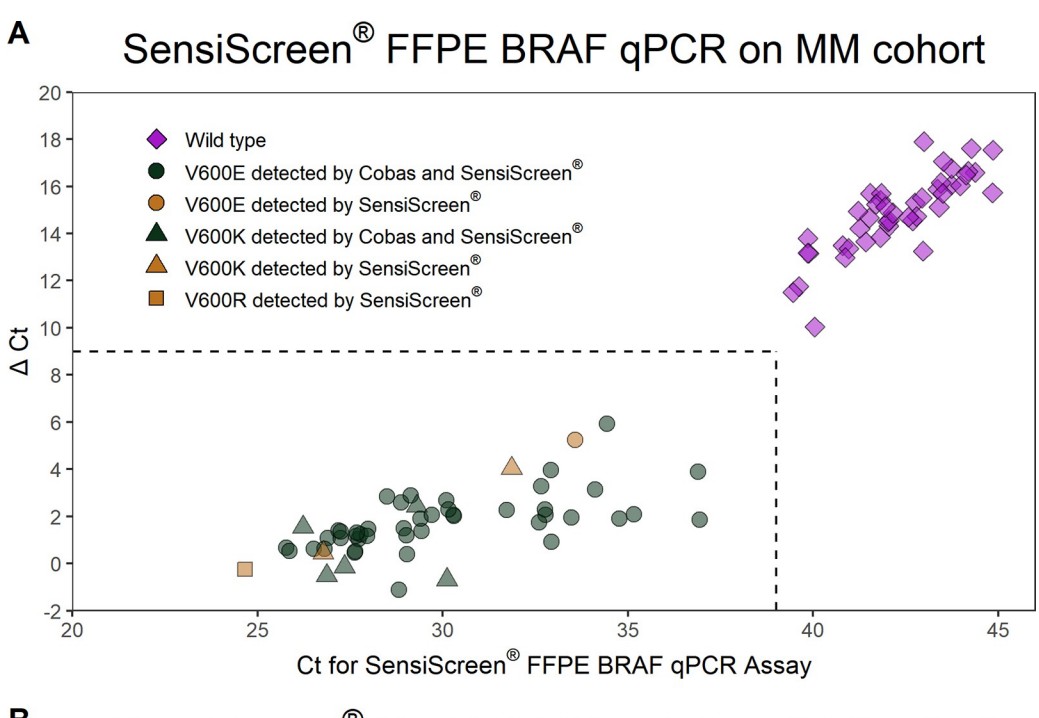

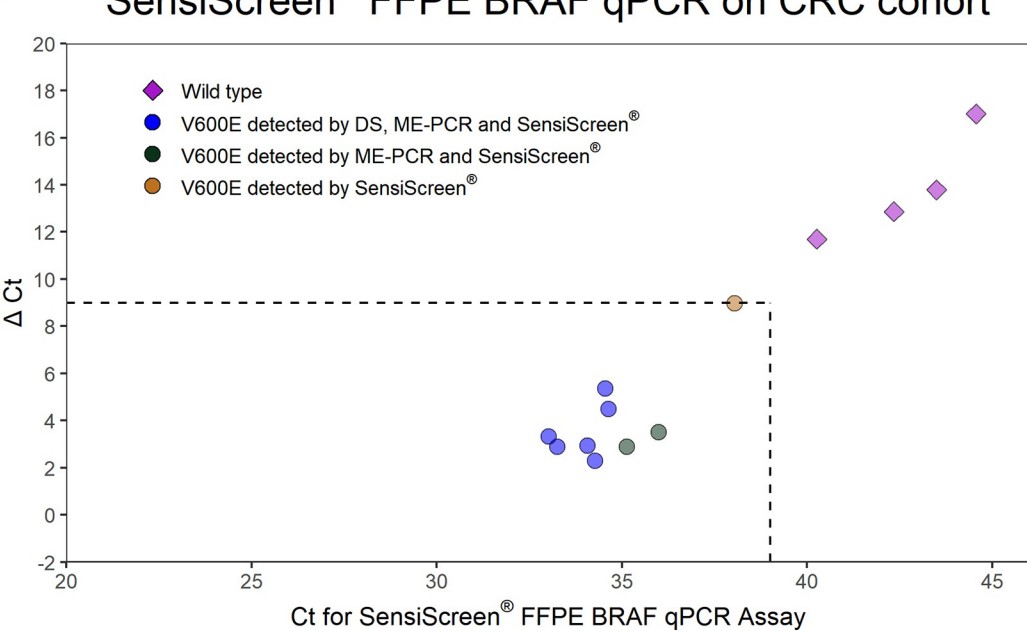

**Fig 1. Dot plot of SensiScreen® FFPE *BRAF* qPCR assay clinical data.** (A) SensiScreen® FFPE BRAF qPCR multiplex assay applied to the MM cohort. BRAF Multiplex positive samples were BRAF V600 mutation subtyped by SensiScreen BRAF V600 simplex. (B) SensiScreen® FFPE BRAF qPCR Multiplex assay applied to the CRC cohort. Samples not resulting in a Ct value is not represented in the figures. The ΔCt ($\leq$ 9) and Ct values ($\leq$ 39) are indicated by the dotted line. Abbreviations; CRC: Colorectal cancer; MM: malignant melanoma; Ct: Threshold cycle.

increase in the technology capability for identification of genetic variants occurring during tumourigenesis will increase our knowledge in the field of molecular biology, allowing in the end a better personalized treatment strategy. Many different BRAF test options are available in the clinical, such as immunohistochemistry, real-time PCR, sequencing, or next-generation

**Table 4. Clinical performance of SensiScreen® FFPE BRAF qPCR assays in comparison to Cobas® 4800, ME-PCR and DS.**

| | Comparison of SensiScreen® to methods for BRAF exon 15 mutation testing | | | | | | | |
|---|---|---|---|---|---|---|---|---|
| | Malignant melanoma FFPE tumours | | | Colorectal cancer FFPE tumours | | | | |
| | SensiScreen® | Cobas 4800[A] | Difference | SensiScreen® V600E | DS | Difference | ME-PCR | Difference |
| WT | 69 | 73 | 4 | 88 | 91 | 3 | 89 | 1 |
| V600D | 0 | 0 | 0 | - | - | - | - | - |
| V600E | 46 | 45 | 1 | 9 | 6 | 3 | 8 | 1 |
| V600K | 7 | 5 | 2 | - | - | - | - | - |
| V600R | 1 | 0 | 1 | - | - | - | - | - |
| Identified mutation | 54 | 50 | 4 | 9 | 6 | 3 | 8 | 1 |
| Total patient samples | 123 | 123 | - | 97 | 97 | - | 97 | - |

The SensiScreen® FFPE BRAF qPCR clinical performance was evaluated on two different FFPE tumour biopsy cohorts, one cohort from MM patients previously analyzed by Cobas® 4800 *BRAF* V600 Mutation Test, and one cohort from CRC patients previously analyzed by ME-PCR and DS. Abbreviations: DS: Direct sequencing, FFPE, formalin-fixed paraffin-embedded; ME-PCR: Mutant enriched PCR, PCR: polymerase chain reaction; WT: wild type.

sequencing (NGS). Direct sequencing is becoming a less commonly used method today, due to its high LOD. The National Comprehensive Cancer Network (NCCN) guidelines recommend BRAF testing in patients where targeted therapy is a possibility. NCCN guidelines does not recommend any specific methodology for BRAF detection [28]. The commercially available qPCR mutations tests, such as Cobas 4800 *BRAF* V600 mutation test, THxID *BRAF* kit (bioMérieux), and Therascreen *BRAF* V600E RGQ PCR kit cannot detect mutations levels below 1%. Further, several methods for *BRAF* mutations test do not discriminate between the V600E/D/K/R variants. A poor LOD may results in false negatives that deprive patients from optimal treatment.

In this work we have demonstrated how our technology makes it possible to discriminate between the V600 variants obtaining results with decreased LOD compared to other assays on the market. The comparison between other methods for *BRAF* mutation testing and the SensiScreen® FFPE BRAF qPCR Assay showed a high concordance and, in addition, demonstrated that the SensiScreen® FFPE BRAF qPCR Assay can identify more *BRAF* mutant cases. Indeed, we found additional mutations on MM patients and on CRC patients when compared to Cobas® 4800 BRAF V600 Mutation Test (Roche), ME-PCR and DS, respectively. Cobas 4800 is, according to the specification, able to detect V600E, V600D and V600K. The SensiScreen® FFPE BRAF qPCR Assay was able to identify one V600R, two V600K and one V600E not detected by Cobas® 4800 BRAF V600. Furthermore, The SensiScreen® FFPE BRAF qPCR Assay was able to identify one V600E not detected with ME-PCR, and three not detected with DS. We have designed and developed a novel BRAF assay for the analysis of DNA extracted from FFPE tissues, possessing high sensitivity and high specificity, which is suitable for use in daily laboratory routine. In addition, the short turnaround time for the SensiScreen® FFPE BRAF qPCR Assay is saving time and makes it attractive for clinical decision making. There is no need of expensive and time-consuming equipment with the requirement of special trained personnel since SensiScreen® FFPE BRAF qPCR Assay is low costs (similar to other PCR based methods)

The developed SensiScreen® FFPE BRAF qPCR Assay is of relevance especially in the case of genes with a hotspot site, such as codon 600 of the BRAF gene. More interestingly, the SensiScreen® FFPE *BRAF* qPCR Assay is able to identify not only the V600E variant, but also other changes that are relevant for MM patients.

In conclusion, the SensiScreen® FFPE *BRAF* qPCR Assay will permit to have a more specific, time and cost saving approach to better characterize, and consequently to better treat,

patients affected by MM, CRC and the other cancer types, for which *BRAF* characterization is fundamental for therapy decision.

## Supporting information

**S1 Fig. Calibration curves of SensiScreen® FFPE BRAF qPCR assay for each target Ct value as function of log (copy number).**
(DOCX)

**S1 Table. Clinical-pathological characteristics of cohort 1 and 2 used for analysis by Cobas® 4800, ME-PCR and DS.** Abbreviations; F: female; M: male.
(DOCX)

## Acknowledgments

The authors declare that there is none to list in the acknowledgments.

## Author Contributions

**Conceptualization:** Jan Lorenzen, Ulf Bech Christensen, Milo Frattini.

**Data curation:** Mariann Guldmann-Christensen, Rasmus Koefoed Petersen, Julie Mejer Holmgaard Duelund, Emma Pipó-Ollé, Samantha Epistolio, Alice Riva, Giulia Dazio, Tine Meyer.

**Formal analysis:** Michael Børgesen, Rasmus Koefoed Petersen.

**Funding acquisition:** Jan Lorenzen, Milo Frattini.

**Investigation:** Mariann Guldmann-Christensen, Julie Mejer Holmgaard Duelund, Majbritt Hauge Kyneb, Emma Pipó-Ollé, Samantha Epistolio, Alice Riva, Giulia Dazio, Tine Meyer.

**Methodology:** Michael Børgesen, Majbritt Hauge Kyneb.

**Project administration:** Jan Lorenzen, Ulf Bech Christensen, Milo Frattini.

**Resources:** Mariann Guldmann-Christensen, Julie Mejer Holmgaard Duelund, Majbritt Hauge Kyneb, Samantha Epistolio, Alice Riva, Giulia Dazio, Elisabetta Merlo, Tine Meyer.

**Supervision:** Milo Frattini.

**Validation:** Anna Lahn Sørensen, Rasmus Koefoed Petersen.

**Visualization:** Anna Lahn Sørensen, Rasmus Koefoed Petersen, Katharina Flugt.

**Writing – original draft:** Anna Lahn Sørensen, Mariann Guldmann-Christensen, Michael Børgesen.

**Writing – review & editing:** Anna Lahn Sørensen, Mariann Guldmann-Christensen, Michael Børgesen, Rasmus Koefoed Petersen, Samantha Epistolio, Giulia Dazio, Milo Frattini.

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
