## [Decision Letter · Decision Letter 0]

14 Nov 2022

PONE-D-22-26405Detection of BRAF mutations in malignant melanoma and colorectal cancer by SensiScreen® FFPE BRAF qPCR assayPLOS ONE

Dear Dr. Sørensen,

Thank you for submitting your manuscript to PLOS ONE. After careful consideration, we feel that it has merit but does not fully meet PLOS ONE’s publication criteria as it currently stands. Therefore, we invite you to submit a revised version of the manuscript that addresses the points raised during the review process.

We look forward to receiving your revised manuscript.

Kind regards,

Alvaro Galli

Academic Editor

PLOS ONE

Journal Requirements:

"ALS, MB, RKP, EPO, KF, MHK, JL and UBC received salaries from Eurostars. ALS, MB, RKP, EPO, KF and UBC are employees of PentaBase A/S. SensiScreen FFPE BRAF qPCR is part of a marketed product portfolio of PentaBase A/S."  

We note that one or more of the authors have an affiliation to the commercial funders of this research study : PentaBase A/S. 

Reviewers' comments:

Reviewer's Responses to Questions

**Comments to the Author**

1. Is the manuscript technically sound, and do the data support the conclusions?

Reviewer #1: Yes

Reviewer #2: No

2. Has the statistical analysis been performed appropriately and rigorously? 

Reviewer #1: I Don't Know

Reviewer #2: Yes

3. Have the authors made all data underlying the findings in their manuscript fully available?

Reviewer #1: Yes

Reviewer #2: Yes

4. Is the manuscript presented in an intelligible fashion and written in standard English?

Reviewer #1: Yes

Reviewer #2: Yes

5. Review Comments to the Author

Reviewer #1: In this study, Authors presented a new methodology based on a quantitative PCR assay for detection of BRAF mutation in melanoma and colorectal cancer. The SensiScreen® FFPE BRAF qPCR Assay here proposed is able to identify not only the V600E variant of BRAF, but also other clinically-relevant mutations in such a gene.

The experimental plan is well developed and description of data is detailed.

My only criticism is the following: it is not appropriate to affirm - into the Discussion section - that the SensiScreen FFPE BRAF qPCR Assay may represent "a better option than NGS, a method characterized by lower sensitivity (5% in fixed tissues) and a longer procedure at least two working days compared to less than 2 hours for the SensiScreen® FFPE BRAF qPCR Assay". The qPCR and NGS approaches ARE TWO DISTINCT MOLECULAR DIAGNOSTIC FIELDS! The comparison of the assay here proposed should be with the other existing quantitative PCR assays (as correctly done into the work). Therefore, this sentence must be deleted since it is misleading.

Reviewer #2: The present study brings another PCR based approach for BRAF mutation detection. It does lack novelty, in a very active market, with point-of care, nanosolutions, etc, and the new methodology is compared with old fashion approaches, such as direct sequencing, which are no longer use in the routine setting.

Extra issues that should be addressed by the authors:

1- Currently the methodologies available with higher sensitive are NGS and ddPCR. Since the aim of the study, as mentioned by the authors, is to “was to validate a highly sensitive, specific, fast, and easy-to-use assay for BRAF”. Most of the advantageous can be found in both NGS and ddPCR. Importantly, It is not anymore acceptable to run against direct sequencing, which has been known for the high LOD rate, and since almost a decade years is not the gold standard for mutation detection. So the comparison made in cohort 2, is meaningless.

2- The introduction most explain the readers the methodology and underlying basis of the PentaBase DNA analogue technology, the core methodology of the manuscript, instead of repletion of BRAF clinical impact that is very long in the introduction and can be synthetize.

In the material and methods:

3-LOB, was not done?

4- “Only patients with sufficient tumour tissue material have been include. Be more precise and rigorous genomic DNA was extracted from six 7 μm thick serial FFPE sections” – this is too much for any current methodology and does not meet the goal of the authors of “SensiScreen® FFPE BRAF qPCR Assay was developed to use a minimal input of DNA.”

5- Clarify - Four patients (cohort 1) and three patients (cohort 2) were excluded due to invalid DNA for the present assay only ? So, the Roche assay and DS had reliable results? How the new assay con overperform the COBAS and DS in these cases?

6- Spell out ME-PCR in MM

7- Discussion should address the price of assays and cost-effective of the methodologies.

6. PLOS authors have the option to publish the peer review history of their article (what does this mean?). If published, this will include your full peer review and any attached files.

Reviewer #1: **Yes: **Palmieri Giuseppe

Reviewer #2: No

---

## [Author Response · Author response to Decision Letter 0]

21 Dec 2022

Dear Reviewers

We thank you for the careful and constructive review of our manuscript, and for the willingness to let us submit a revision of the manuscript for review in PLOS ONE. The comments have been very helpful and have greatly improved the manuscript.

We have addressed the comments and the manuscript has been revised accordingly. 

Best regards

Anna Lahn Sørensen, PhD

---

## [Decision Letter · Decision Letter 1]

26 Jan 2023

Detection of BRAF mutations in malignant melanoma and colorectal cancer by SensiScreen® FFPE BRAF qPCR assay

PONE-D-22-26405R1

Dear Dr. Sørensen,

We’re pleased to inform you that your manuscript has been judged scientifically suitable for publication and will be formally accepted for publication once it meets all outstanding technical requirements.

Kind regards,

Alvaro Galli

Academic Editor

PLOS ONE

Additional Editor Comments (optional):

Reviewers' comments:

Reviewer's Responses to Questions

**Comments to the Author**

1. If the authors have adequately addressed your comments raised in a previous round of review and you feel that this manuscript is now acceptable for publication, you may indicate that here to bypass the “Comments to the Author” section, enter your conflict of interest statement in the “Confidential to Editor” section, and submit your "Accept" recommendation.

Reviewer #1: All comments have been addressed

2. Is the manuscript technically sound, and do the data support the conclusions?

Reviewer #1: Yes

3. Has the statistical analysis been performed appropriately and rigorously? 

Reviewer #1: Yes

4. Have the authors made all data underlying the findings in their manuscript fully available?

Reviewer #1: Yes

5. Is the manuscript presented in an intelligible fashion and written in standard English?

Reviewer #1: Yes

6. Review Comments to the Author

Reviewer #1: Authors have somehow addressed all main raised issues, rendering the manuscript much more useful to readers.

7. PLOS authors have the option to publish the peer review history of their article (what does this mean?). If published, this will include your full peer review and any attached files.

Reviewer #1: No

---

## [Editor Report · Acceptance letter]

1 Feb 2023

PONE-D-22-26405R1 

Detection of BRAF mutations in malignant melanoma and colorectal cancer by SensiScreen FFPE BRAF qPCR assay 

Dear Dr. Sørensen:

I'm pleased to inform you that your manuscript has been deemed suitable for publication in PLOS ONE. Congratulations! Your manuscript is now with our production department. 

Kind regards, 

on behalf of

Dr. Alvaro Galli 

Academic Editor

PLOS ONE